# Nutritional and Bioactive Compounds in Mexican Lupin Beans Species: A Mini-Review

**DOI:** 10.3390/nu11081785

**Published:** 2019-08-02

**Authors:** Mario Alberto Ruiz-López, Lucia Barrientos-Ramírez, Pedro Macedonio García-López, Elia Herminia Valdés-Miramontes, Juan Francisco Zamora-Natera, Ramón Rodríguez-Macias, Eduardo Salcedo-Pérez, Jacinto Bañuelos-Pineda, J. Jesús Vargas-Radillo

**Affiliations:** 1Botany and Zoology Department, CUCBA, University of Guadalajara, Zapopan, Jalisco ZP 45110, Mexico; 2Wood, Pulp and Paper Department, CUCEI, University of Guadalajara, Zapopan, Jalisco ZP 45110, Mexico; 3Research in Behavioral Feeding and Nutrition Institute CUSUR, University of Guadalajara, Cd. Guzman, Jalisco ZP 49000, Mexico; 4Veterinary Medicine Departarment CUCBA, University of Guadalajara, Zapopan, Jalisco ZP 45110, Mexico

**Keywords:** legumes, Mexican lupins, protein isolates, dietary fiber, nutrients, bioactives compounds

## Abstract

As a source of bioactive compounds, species of the genus *Lupinus* are interesting legumes from a nutritional point of view. Although wild species are abundant and represent a potential source of nutrients and biologically active compounds, most research has focused on domesticated and semi-domesticated species, such as *Lupinus angustifolius*, *Lupinus albus*, *Lupinus luteus*, and *Lupinus mutabilis*. Therefore, in this review, we focus on recent research conducted on the wild *Lupinus* species of Mexico. The nutritional content of these species is characterized (similar to those of the domesticated species), including proteins (isolates), lipids, minerals, dietary fiber, and bioactive compounds, such as oligosaccharides, flavonoids, and alkaloids.

## 1. Introduction

At present, epidemiological studies have shown that plant-based diets reduce the risks of developing chronic diseases [1,2]. In this sense, legumes represent one of the most important food categories. They have been widely cultivated and used as basic food sources to cover protein and energy needs throughout human history [3]. Legume seeds are used as a source of proteins, lipids, and dietary fiber in human and animal nutrition, and they are also adapted to marginal soils and climates. In many regions of the world, legumes are the only source of dietary protein and have been used as an effective substitute for animal protein at a lower cost.

Additionally, it has been proven that legumes contain a large number of bioactive compounds with potential health benefits, such as the prevention of coronary heart disease, cancer, and diabetes. Among these compounds are oligosaccharides, phenols, and alkaloids. The interest in these compounds is due to their possible beneficial applications (Figure 1) as metabolic, hormonal, and digestive system regulators, as well as prebiotics [4], so it is of great interest to increase the knowledge of legumes as functional foods and the importance of their bioactive compounds [2].

## 2. Domesticated Lupins

Despite their high protein and dietary fiber content and potential health benefits, lupins are undervalued legumes. There is a great diversity of species, but few are cultivated. *Lupinus angustifolius*, *Lupinus albus*, *Lupinus luteus*, and *Lupinus mutabilis* (Figure 2, Figure 3, Figure 4 and Figure 5) are among those that are domesticated. Domesticated species have been improved and have (a) low alkaloid content to increase edibility for humans and animals, (b) soft seeds with high germination, and (c) indehiscent pods that keep their seeds and yield efficient harvests [5].

In relation to their nutritional properties, the protein content in lupine seeds is similar to that in soy (Table 1), with an acceptable content of essential amino acids. In addition, the presence of prolamins and low glutelin content makes lupine proteins desirable for the preparation of foods low in gluten. A gluten-low diet is one that excludes most grains and is recommended for people who have celiac disease or gluten sensitivity. For other people, however, going gluten-free can be unhealthy. The benefits and risks of a gluten-free diet should be carefully weighed, especially if the person starting the new diet does not truly need to restrict gluten intake.

The protein fractions of several species of lupines have been isolated, and the two most important are albumins and globulins, which are present at a ratio of 1:9. Globulins represent about 90% of the total protein content in the seeds, and they can be separated by ultracentrifugation and chromatography into two main components, vicilins and legumins (also called α-conglutin and β-conglutin), and two minor proteins, γ-conglutin and δ-conglutin [8].

*Lupinus albus* γ-conglutin in vitro models have demonstrated the biological activity that potentiates insulin and metformin activity on cellular glucose consumption, thereby presenting a potential use for γ-conglutin in glycemic control [9]. Furthermore, protein isolates of *L. albus* have been reported to have a hypolipidemic, anti-atherosclerotic, and hypocholesterolemia effect in rabbits, rats, and chickens, and were shown to increase LDL receptor activity in HepG2 cells [10,11,12].

Several clinical human studies have shown that the incorporation of 25 g of lupine protein into different foods decreases total LDL and HDL cholesterol, as well as triglycerides and uric acid, in hypercholesterolemic subjects [13].

Lupine seeds are an important source of nutritionally important lipids. Table 2 shows the average composition of oils in the seeds of different lupine species.

The content of essential fatty acids, such as linoleic acid (18:2), is higher in *L. luteus* than in the other two species in Table 2. However, *L. albus* has a higher content of linolenic acid (18:3). Similarly, the fatty acid composition of *L. angustifolius* seeds is excellent because of its high palmitic and stearic acid content, which makes it an attractive ingredient for food products and nutraceuticals [14].

The lipid content of *L. albus* is particularly promising since its composition is very close to the dietary recommendations for the prevention of cardiovascular diseases. Among these recommendations is the intake of similar proportions of omega-6 and omega-3 fatty acids, while the diet of Western countries has a large omega-6/omega-3 ratio of 15:1. Excess omega-6 relative to omega-3 represents a risk factor, whereas a 2:1 ratio has a positive effect and reduces the mortality associated with cardiovascular disease [15].

The beneficial effect of dietary fiber consumption on the digestive system is known, especially for the prevention of colon cancer. According to the Nordic Nutrition recommendations, an adequate dietary fiber consumption reduces the risk of constipation, which, in turn, reduces the risk of colorectal cancer and other chronic diseases, such as cardiovascular disease and diabetes. Thus, the consumption of foods rich in fiber is favorable to health. A dietary fiber intake of 25–35 g per day is recommended.

*Lupinus angustifolius* is a rich source of dietary fiber (41.5%), of which 11% is soluble fiber and 30.5% is insoluble fiber [16]. In *L. albus*, a content of 50.4% total dietary fiber has been reported, with 2.0% soluble fiber and 48.4% insoluble fiber. Fiber incorporated into food products can benefit intestinal functions and fecal parameters by decreasing the risk of colon cancer and promoting a healthy digestive system [17,18].

Lupine seeds are a good source of minerals, such as calcium, phosphorus, and iron, among others, which are important for various bodily functions and are part of many tissue structures. The most important minerals in human nutrition are calcium, phosphorus, iron, iodine, fluorine, and zinc.

Lupines, similar to other legumes, are generally a good source of minerals (Table 3), the content of which varies among species and differs from that of other legumes [19]. Compared with other legumes, lupines have low levels of calcium (Ca) and phosphorus (P) but similar contents of microelements, such as iron (Fe), zinc (Zn), and copper (Cu) [20]. *Lupinus angustifolius* var. troll presents a higher Ca content, while *L. luteus* var. 4492 contains more Mg, P, Cu, Fe, and Zn [21].

## 3. Bioactive Compounds

Additionally, lupines contain bioactive compounds, such as oligosaccharides, phenolic compounds, and alkaloids, some of the main biomolecules that can prevent and protect against chronic diseases, such as cancer, diabetes, and neurodegenerative and cardiovascular diseases [22,23].

The beneficial effect of these compounds depends on their chemical structures, concentrations, times of exposure, interactions with other compounds, and, especially, their bioavailabilities [24,25]. These compounds are considered to be antinutrients and/or pronutrients with negative and/or positive effects on health [23,26]. Thus, it is important to know the quantity and type of compounds in food and understand how they affect the organism.

On the other hand, most of the oligosaccharides of plant origin are α-galactosides and belong to the raffinose family [27], which includes raffinose (a trisaccharide), stachyose (a tetrasaccharide), verbascose (a pentasaccharide), and ajugose (a hexasaccharide) (Figure 6).

In domesticated lupine cultivars and species, there is a variation in the content of total oligosaccharides, with a range from 5.46 (in *L. albus*) to 12.3 g/100 g for dry samples (in *L. luteus*). Stachyose is the main oligosaccharide, with values that range from 3.62 to 8.61 g/100 g for dry sample (Table 4).

Similarly, phenolic compounds are bioactive secondary metabolites that are found mainly in plants. They have anti-allergenic, anti-arteriogenic, anti-inflammatory, antimicrobial, antioxidant, anti-thrombotic, cardioprotective, and vasodilator effects [29]. There is variation in the content of phenolic compounds among lupine species, as well as among cultivars, growing sites, and parts of the plant. The total phenol varies from 212 to 317 mg per 100 g DM in the seeds of cultivars of *L. albus*, *L. luteus*, and *L. angustifolius* [30], but these values are inferior to leaves of traditional medicinal plants used in China, India, and Europa (*Pimpinella anisum*), with values of 234 to 1178.5 mg GAE/100 g DM [31,32,33]. 

*Lupinus albus* was revealed to have the highest total flavonoid content, i.e., 1100 μg catechin/g DM, compared with varieties of *L. angustifolius* (Table 5) [34].

The majority of phenolic compounds in lupine species are flavones, dihydroflavones, phenolic acids, and isoflavones (Table 6).

The varieties of *L. luteus* have the highest content of apigenin-6,8-di-C-b-glucopyranoside and protocatechuic acid compared with the varieties of the other two species, while the main phenolic acid in *L. albus* is p-hydroxybenzoic.

In *L. angustifolius* seeds, a higher concentration of p-hydroxybenzoic acid was observed. Additionally, in other studies, flavones (mainly luteolin glycosides, apigenin, and diosmetin), isoflavones (mainly derived from genistein), and dihydroflavones have been reported in this species [35].

Many of these phenolic acids and flavonoids are known for their high antioxidant capacity, which has been the subject of several studies aiming to understand this property, as seen in Table 7.

It can be observed that the highest antioxidant capacity has been found in *L. luteus* varieties using the DPPH technique (2,2-diphenyl-1-picrilhydrazil), while varieties of *L. angustifolius* have shown the highest antioxidant activity with the TRAP technique (total peroxil radical-trapping potential).

## 4. Mexican Wild *Lupinus*

At the moment, diverse investigations focusing on wild and underused legumes are underway. It is known that wild legumes, such as *Lupinus* spp., have significant quantities of proteins, essential amino acids, polyunsaturated fatty acids (PUFAs), dietary fiber, minerals, and essential vitamins, comparable to edible legumes, in addition to the presence of beneficial bioactive compounds [36].

The study of Mexican lupine species is very recent. Dr. Ruiz-López started to study these species in 2000 by describing their distribution in Mexico, analyzing their chemical-nutritional and antinutrient composition, and obtaining their protein isolates [37,38].

### 4.1. Distribution in Mexico

Because of the biological, geographic, and climatic diversity in Mexico, nearly 100 wild species have been identified and are distributed throughout the country. The highest concentration of these species is found in the Sierra Madre Occidental and the Transverse Neovolcanic Axis [39], mainly in areas with an altitude between 1500 and 3500 m above sea level (Figure 7, Figure 8, Figure 9 and Figure 10).

### 4.2. Nutritional Composition

The wild lupines of Mexico present nutritional values similar to those of domesticated species.

The protein content (37.2–45.4 g 100 g^−1^) (Table 8) is higher than that of other wild legumes, such as *Caesalpinia bracteosa* (29.3 g 100 g^−1^ DM), *Dimorphandra gardneriana* (32.3 g 100 g^−1^ DM), *Pterogyne nitens* (28.3 g 100 g^−1^ DM), *Hymenaea courbaril* (10.9 g 100 g^−1^ DM), *Senna obtusifolia* (22.3 g 100 g^−1^ DM), and *Senna rugosa* (22.3 g 100 g^−1^ DM) [37], and similar to that of domesticated lupine species.

### 4.3. Protein Isolates and Their Functionality

The nutritional value of food proteins is determined by their quality, which depends on their composition and the biological availability of their essential amino acids to humans: phenylalanine, isoleucine, leucine, lysine, methionine, threonine, tryptophan, valine, and histidine. Thus, protein isolates of wild lupins from Mexico have been prepared, as can be observed in Table 9.

In Mexican species, protein isolates of 93–95% protein were obtained by isoelectric precipitation. The meal was suspended in water (10% *w*/*v*) and the pH was adjusted to 9.0. It was centrifugated, and the suspension was adjusted to pH 4.5 for protein precipitation. Then, the suspension was left at 4 °C overnight to allow the proteins to precipitate. Then, centrifugation was performed at 10,000× *g* for 10 min at 4 °C. The protein precipitate was freeze dried in a Freezone Dry System (Labconco).

In addition, protein isolates from wild Mexican species have α, β, and γ conglutin, which may be promising as hypoglycemic, hypolipidemic, anti-atherosclerotic, and hypocholesterolemic agents. Fatty acids are the organic components of fats and are crucial because they supply energy to our body and enable the healthy formation and maintenance of different tissues.

The fatty acid content was determined in Mexican species, which have a high lipid content of up to 13%. Table 10 shows the composition of fatty acids in three lupine species from different localities in Mexico. Fatty acid methyl esters (FAME) were prepared as follows: lupin oil was placed into a glass tube with a screw-cap. The tubes were flushed with nitrogen until dry. After cooling, 2 mL of hexane was added to each tube, re-capped, mixed, and left standing to allow the hexane layer to separate. 1 mL of each hexane layer was diluted 10-fold with hexane, and 1 mL was transferred to autosampler vials.

The analysis of FAME was performed using a Perkin-Elmer Autosystem 1-A gas chromatograph equipped with a flame ionization detector. FAME were identified by comparing their retention times with known concentrations of methyl ester standards, including nonadecanoic methyl esters (Sigma Chemical Co. Ltd, St Louis, MO, USA) that were analyzed under similar conditions.

There have been few reports on the fatty acid content of Mexican lupines, with only one report on three species. In that study, it was observed that the fatty acid profile of the three Mexican species presented a predominance of unsaturated fatty acids (USFAs). Those existing in higher quantities are linoleic acid (C18:2), followed by oleic acid (C18:1), and gamma-linolenic acid (C18:3). The undesirable erucic acid (C22:1) was not found in any of the species studied [47].

Essential fatty acids, such as linoleic acid, are higher in *Lupinus exaltatus* than in all wild species of lupine; in *Lupinus montanus* and *Lupinus exaltatus*, linoleic acid was higher than in *L. angustifolius* and *L. albus*, respectively [47]. Similarly, *L. montanus* and *L. stipulatus* had higher linolenic acid content than domesticated lupine species, and *L. exaltatus* had a greater concentration of this fatty acid than *L. angustifolius*. The n6/n3 ratio was lower in *L. exaltatus* than in domesticated lupines analyzed for this trait, and this ratio was lower in *L. montanus* and *L. stipulatus* than in *L. albus*. These results are interesting given the recommended intake of similar amounts of omega-6 and omega-3 fatty acids for the prevention of cardiovascular diseases. Moreover, a good ratio of unsaturated vs. saturated fatty acids is observed, which could be beneficial for the development and maintenance of brain functions, as well as immune and inflammatory responses. Mammalian cells cannot convert omega-6 to omega-3 fatty acids because they lack the converting enzyme, omega-3 desaturase [15].

### 4.4. Dietary Fiber

Dietary fiber consumption is critical. Several studies have shown that fiber reduces blood pressure in patients with hypertension and also suggest a small reduction in normotensive ones [48].

The content of dietary fiber (DF) in the seeds of Mexican species (Table 11) varies considerably, with values ranging from 17.72 (*L. exaltatus*) to 27.93 g/100 g (*L. rotundiflorus*) [49]. These values are lower than those of domesticated species, such as *L. albus* with 50.4 g/100 g and *L. angustifolius* with 41.5 g/100 g [16,18]. However, the DF values of lupines are higher than those reported in other legumes for human consumption, such as soybean, beans, and mung beans.

### 4.5. Minerals and Bioavailability

In Mexican lupine seeds, the mineral content has been evaluated, as shown in the following table. Mineral concentration is highly variable, depending on the species (Table 12). *Lupinus mexicanus* has the highest values of Ca (3252 mg/kg), while *L. montanus* has the highest P content, with 6500–7690 mg/kg. *L. rotundiflorus* has the highest Fe content with 82.8 mg/kg, and *L. exaltatus* has the highest Zn and Cu content recorded, with values of 89.6 and 184 mg/kg, respectively.

The values of Ca in *L. mexicanus*, *L. montanus*, and *L. rotundiflorus* are higher than those reported in varieties of domesticated species of lupine, such as *L. angustifolius* (1430–1618 mg/kg), *L. albus* (1336–1390 mg/kg), and *L. luteus* (1104–1348 mg/kg). The values of P in *L. montanus* are similar to those found in varieties of *L. luteus* (7155–8457 mg/kg) and higher than those of *L. albus* and *L. angustifolius. Lupinus rotundiflorus* and *L. montanus* have a higher Fe content than domesticated lupines, since they contain values ranging from 38 to 70 mg/kg, which makes them an interesting source of Fe.

Fe is one of the most important nutritional minerals because its deficiency is associated with anemia. Thus, in a study on *Lupinus rotundiflorus*, the bioavailability of Fe was evaluated using a rat model. It was found that the concentration of Fe was higher in roots than in cooked seeds, and both showed good bioavailability, with 13.8% and 13.7%, respectively, compared to iron sulfate (18.38%). Thus, this species could be considered as a source of bioavailable Fe and incorporated into foods [51].

## 5. Bioactive Compounds

### 5.1. Oligosaccharides

The raffinose family of oligosaccharides (RFOs) are indigestible carbohydrates (α-galactosides) in the intestinal tract. They are fermented in the colon and selectively stimulate the establishment and development of beneficial bacteria, such as *Bifidobacterium*, which have beneficial effects on the health of the host, including improvement in mineral absorption, modulation of lipid levels, reduction of colon cancer risk, influence on glucose levels, reduction and prevention of intestinal infections, and stimulation of the immune system [52].

Table 13 shows the content of oligosaccharides in Mexican wild species, obtained by using 50% ethanol twice. The obtained extract was boiled under reflux for 10 min. The precipitated solids were removed by centrifugation, and then the solids were concentrated under a vacuum to subsequently precipitate the oligosaccharides with 100% ethanol. These oligosaccharides were separated by centrifugation, and the ethanol was removed by vacuum. In the purification process, the sample was resuspended in 20 mL of water and filtered through 300 mL of a 1:1 (*w*/*w*) mixture of soil of diatoms and activated carbon. Oligosaccharides recovered with 500 mL of ethanol at 70% (*v*/*v*). Oligosaccharideswas finally eluted by a column DOWEX 50WX8 cation exchange [53] and analyzed using HPLC-RID equipment.

Mexican lupine species have a good total oligosaccharide content, especially *Lupinus campestris*, with 90.26 mg/g per sample, which is greater than the rest of the wild species (Table 13) and similar to *L. albus* varieties (5.46–8.51 g/100 g), *L. angustifolius* (5.3–8.82 g/100 g), and *L. luteus* varieties (9.46–12.3 g/100 g). The sucrose level in *L. campestris* is higher than that in varieties of *L. luteus* and some varieties of *L. albus* but lower than varieties of *L. angustifolius* (2.55–5.05 g/100 g). *Lupinus montanus*, with the highest raffinose content of all wild species (14.55 mg/g), also contains more of this oligosaccharide compared with domesticated lupine varieties, except for *L. albus* varieties. In general, Mexican species show lower stachyose values compared to domesticated lupine varieties, but their verbascose content is higher.

### 5.2. Phenols

Most phenolic compounds are considered to potentially benefit health, for example, by reducing the risks of cardiovascular and neurodegenerative disease, as well as cancer, diabetes, and osteoporosis [25]. In addition, polyphenols are the most attractive natural antioxidants, as they are associated with preventing degenerative diseases, such as cancer and atherosclerosis, as well as for the nutraceutical, cosmetic, and pharmaceutical industries, because they can replace synthetic antioxidants [56].

Few studies have been conducted on the phenol content of Mexican lupine species. For the extraction of flavonoids, the samples were homogenized in 80% MeOH and placed in an ultrasonic bath for 30 min. Subsequently, the bath was filtered with a vacuum pump and centrifuged at 14,000 rpm for 5 min. Finally, the bath was decanted and the supernatant was freeze-dried in a lyophilizer (LAB CONCO Corporation, Kansas City, MI, USA). The lyophilized supernatant was kept at −80 °C until its use in analysis by LC-MS techniques. It reported a predominance of glycosylated and acidulated flavonoids of genistein, isoflavone, flavone, and flavonol in the leaves and roots of nine wild lupine species: *Lupinus reflexus*, *Lupinus elegans*, *L. exaltatus*, *Lupinus hintonii*, *L. mexicanus*, *L. montanus*, *L. rotundiflorus*, *L. stipulatus*, and *Lupinus* sp. Among these species, the following phenols have been found: genistein’s glycated and acidulated flavonoids, such as 2′-hydroxygenistein 4′,7-O-diglucoside, 2′-hydroxygenistein 7-O-glucosylglucoside, genistein 4′,7-O-diglucoside, 2′-hydroxygenistein C,O-diglucoside, 2′-hydroxygenistein 4′,7-O-diglucoside malylated, genistein 6,8-C-diglucoside, 2′-hydroxygenistein 4′,7-O-diglucoside malonylated, and luteone 4′,7-O-diglucoside [57]. Wojakowska et al. (2013) [58] reported the isoflavones genistein, 2′hydroxygenistein, luteolin, wighteona; the flavones acacetin, apigenin, chrysoeriol, luteolin; the flavonols isorhamnetin, kaempferol, quercetin; and the flavanones eriodictyol and naringerina.

The flavonoids present in wild lupins are important. These flavonoids include the flavones (acacetin, apigenin, chrysoeriol, luteolin), flavonols (isorhamnetin, kaempferol, quercetin), and flavanols (quercetin, kaempferol) (Figure 11), since they have been shown to reduce coronary heart disease and cancer and trap reactive oxygen and nitrogen species, which is why they are potent antioxidants [56].

No analysis of the content of phenolic acids or anthocyanins in these species has been conducted, so information is lacking regarding the potential for bioactive phenolic compounds with antioxidant properties, among others.

In seeds of *L. campestris*, a study was conducted to evaluate the effect of germination in their antioxidant capacity, using the DPPH method, and an increase of 51%– 585% was observed only in the second germination day but decreased on the ninth germination day (a decrease of 38%) [55].

The antioxidant activity depends on the type, category, and abundance of these compounds, which have substantial impacts and potential health benefits [33]. It has been reported that phenolic compounds are the main metabolites that have a positive effect on health, largely due to their antioxidant properties, through their reduction power, their ability to eliminate and inhibit free radicals, and their metal chelating activity, which avoids DNA damage and lipid peroxidation [59].

### 5.3. Alkaloids

In general, most of the alkaloids present in the seeds of edible legumes have been reported in species of the genus *Lupinus*. However, sweet varieties have been generated by selection and improvement, leading to low alkaloid content in species to be used in human and animal food [60]. Wild lupine species are rich in quinolizidinic alkaloids (more than 150 different quinolizidine alkaloids have been reported), and their content and composition depends on the phenology, organ of the plant (leaves, stems, flowers, roots, and seeds), as well as their environmental conditions and interactions with herbivores and phytopathogenic microorganisms. Seeds contain the highest total alkaloid content, which varies from 1.5% to 4%, depending on the species [40].

The alkaloid extraction was performed as follows: One half gram of seed flour was homogenized in 5% trichloroacetic acid (3 × 5 mL) and centrifuged at 3000 r.p.m. for 5 min. Later, 10 M NaOH was added to the supernatant, and the alkaloids were extracted with dichloromethane (3 × 5 mL). The dichloromethane was evaporated, and the alkaloids were dissolved in methanol [61].

Table 14 shows that sparteine was either not abundant or not present in the species studied, with the exception of *L. montanus* and *L. reflexus*, which had 3.97 and 26.63 mg of alkaloids/g, respectively, in the sample. Lupanine was present in all samples analyzed, especially in *L. mexicanus*, with 5.05–21.2 mg of alkaloids/g (Figure 12). Additionally, gas chromatography/mass spectrometry (GLC/MS) was used to identify the following alkaloids in *L. campestris*: Aphyllidine, 5,6-dehydrolupanine, Aphylline, Lupanine, a-lsolupanine, and Hydroxyaphyllidine, which was the main alkaloid compound [62]. Similar testing on *Lupinus aschenbornii* identified angustifoline, esparteine, tetrahydrorombifoline, lupanine, multiflorane, and 17-Oxolupanine [63].

The presence of quinolizidinic alkaloids is an advantage of wild lupines over domesticated lupines since these compounds have pharmacological properties. For example, sparteine is used in the treatment of cardiac arrhythmias and induces uterine contractions. In addition, it has been shown to have depressant effects on the central nervous system and hypotensive, diuretic, and anti-inflammatory activities [65,66]. Similarly, lupanine, 13-hydroxylupanine, and multifluorine have pharmacological activities as anticonvulsant, antipyretic, and hypoglycemic agents [67,68,69,70].

In this sense, *L. reflexus* represents an excellent source of esparteine, even greater than that of *Cytisus scoparius*, which is the commercial source of pharmacologically used sparteine. Further, the species *L. mexicanus*, *L. rotundiflorus*, and *Lupinus madrensis* have shown high lupanine concentrations, which could be used in various treatments.

Unlike chemical drugs that have negative physical and sometimes mental side effects, treatments with medicinal plants have shown minimal or no side effects. Consequently, there is currently a tendency to use plants with medicinal properties as food ingredients for the prevention of diseases [59].

In Mexico, there are close to 100 wild species of *Lupinus* widely distributed. These species show great potential due to their high protein content, which varies from 30 to 43 g/100 g, as well as their oil content, which ranges from 5.8 to 13 g/100 g, depending on the species, variety, and environmental conditions.

The wild lupins under consideration represent a potential protein supply and could be domesticated and used for feed if the alkaloids were eliminated and the protein were supplemented with methionine, or if the lupins were used in a mixture with cereals (as an addition to popular foods, such as breads, cookies, salads, etc.). 

The wild Mexican lupin has high potential nutritional value for its high protein, essential fatty acids, essential minerals (such as iron with high bioavailability), and low gluten content. This species is also a future prospective nutraceutical because it contains bioactive compounds, such as a prebiotic of oligosaccharides that improves mineral absorption, the modulation of lipid levels, and reduces colon cancer risk, among other benefits. In Mexican lupins, polyphenols, such as genistein, flavonoids, flavones, and flavolos, show significant potential human health benefits. For example, they can serve as natural antioxidants by acting as scavengers of free radicals, thereby inhibiting lipid peroxidation and acting as agents for chelating metal ions. These benefits have attracted marked attention from the nutraceutical and cosmetic industries, and also from preventive medicine because of their potential application in making food, cosmetic, and pharmaceutical products that can replace synthetic antioxidants associated with health concerns, such as putative carcinogenic effects [13]. Furthermore, alkaloids, such as spartein, lupanin, 13-hydroxylupanine, and multifluorine, which are very abundant in wild species, have pharmacological activities in the central nervous system and a hypoglycemic effect on the pharmaceutic sector. However, it is very important that the molecules do not change when ingested, because their effect could be different, as in the alkaloid Frangufoline, a sedative cyclopeptide alkaloid that does not change in the gastric juices, so it is absorbed unchanged and performs its function [71].

## 6. Conclusions

Mexican wild lupines have substantial uses because they are an important source of protein with an adequate balance of amino acids, which can be extracted and used as ingredients in the preparation of various foods. They have the advantage of low gluten content and are also important sources of dietary fiber, minerals, and essential lipids. Additionally, they contain bioactive compounds, such as oligosaccharides with prebiotic functions, polyphenols, and alkaloids, which have been shown to have various physiological–metabolic properties. Wild lupine species have not yet been the subject of adequate study. However, these species represent a potentially important future nutritional, nutraceutical, and pharmacological source. Thus, it would be interesting to further expand the present knowledge of these species, including those outside of Mexico, since there is a great diversity of species around the world.

## Figures and Tables

**Figure 1 nutrients-11-01785-f001:**
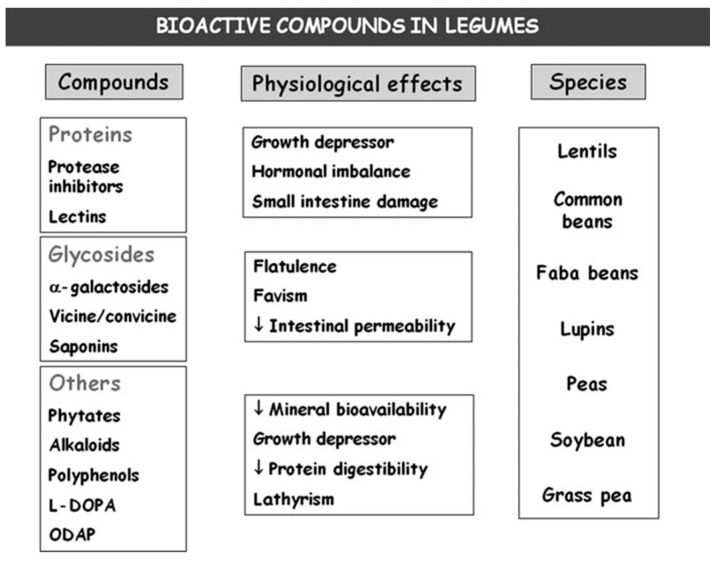
Bioactive compounds in legume seeds and their and physiological effects (adapted from Muzquiz et al., 2012 [4]). L-DOPA (L-Dopaminie), ODAP (oxalyl-L-α,β-diaminopropionic acid).

**Figure 2 nutrients-11-01785-f002:**
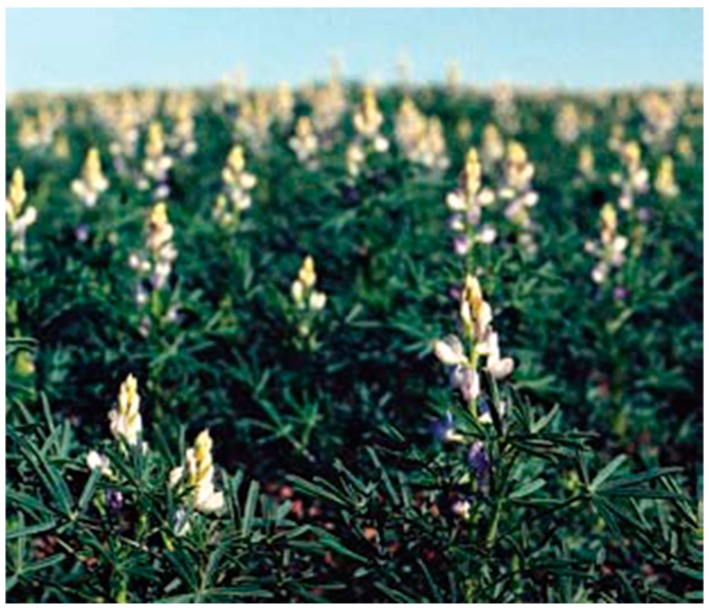
*Lupinus angusifolius* (narrow-leafed lupin), Photo: P. White.

**Figure 3 nutrients-11-01785-f003:**
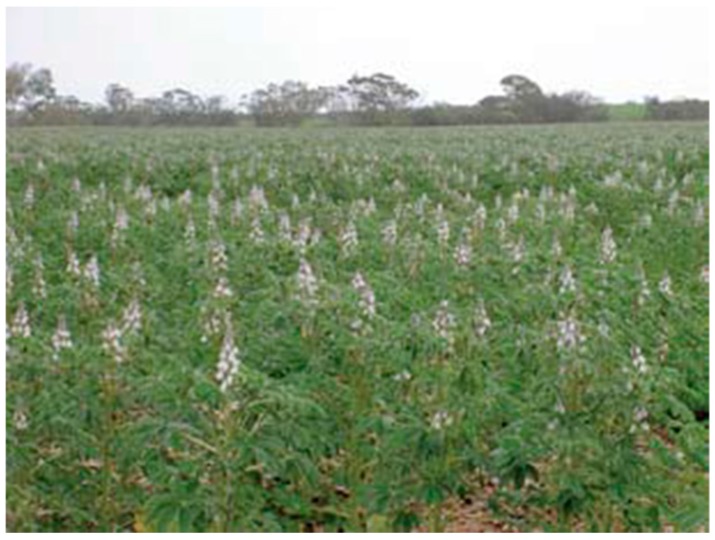
*Lupinus albus* (withe lupin) Photo: B.Buirchell.

**Figure 4 nutrients-11-01785-f004:**
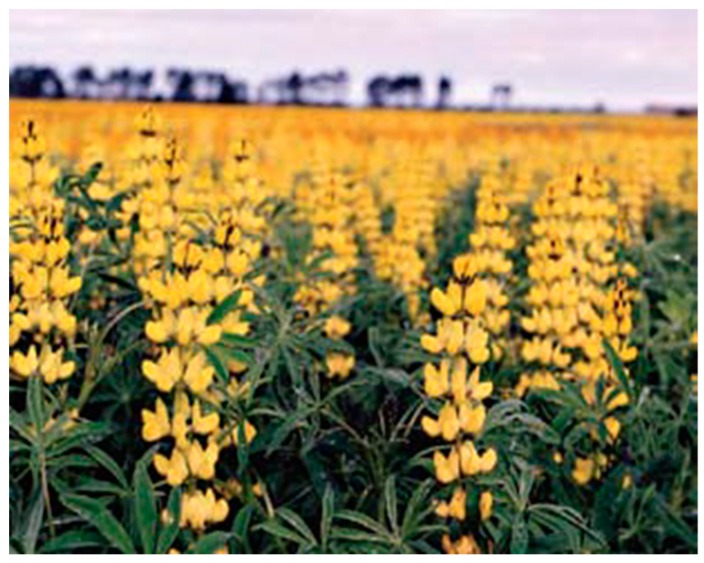
*Lupinus luteus* (Yellow lupin) Photo: P. White.

**Figure 5 nutrients-11-01785-f005:**
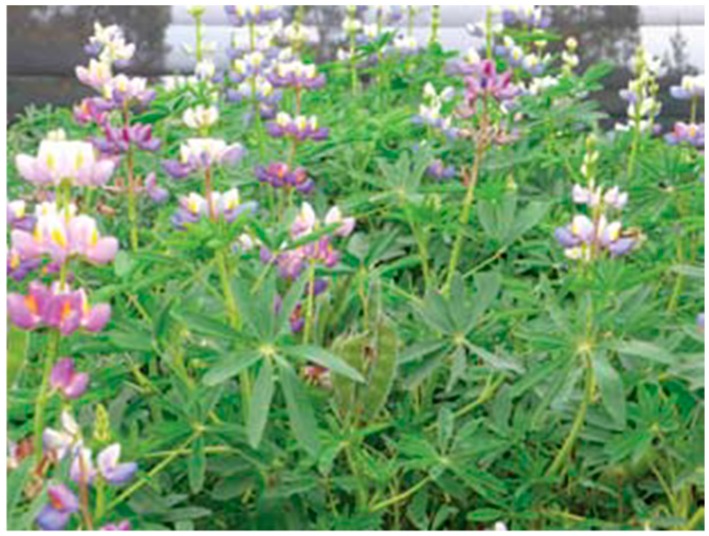
*Lupinus mutabilis* (pearl lupin) Photo: P. White.

**Figure 6 nutrients-11-01785-f006:**
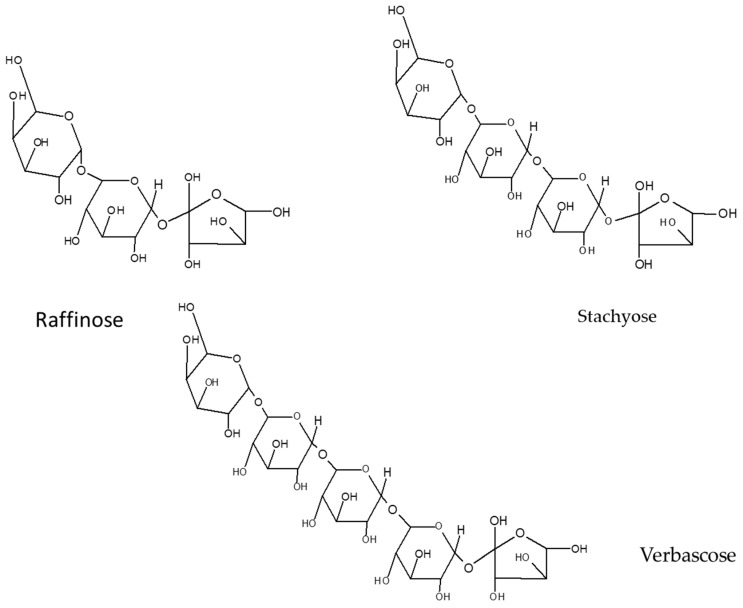
Raffinose family oligosaccharides structures.

**Figure 7 nutrients-11-01785-f007:**
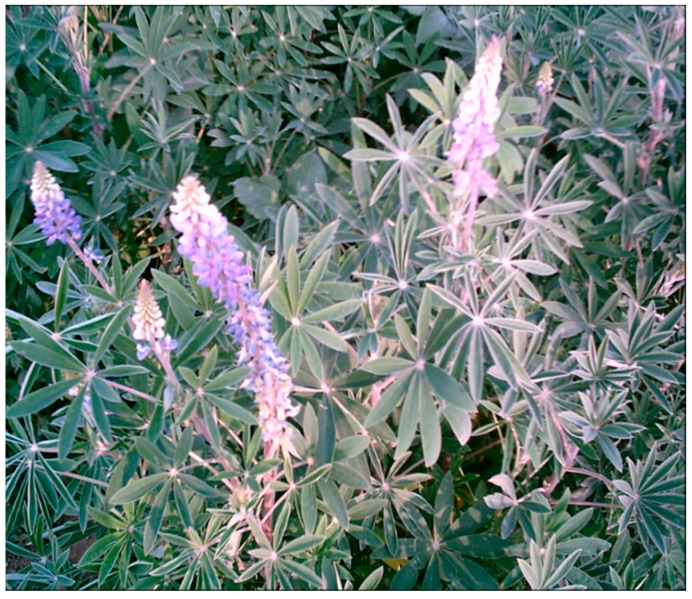
*Lupinus exaltatus* Zucc. Photo: M. Ruiz.

**Figure 8 nutrients-11-01785-f008:**
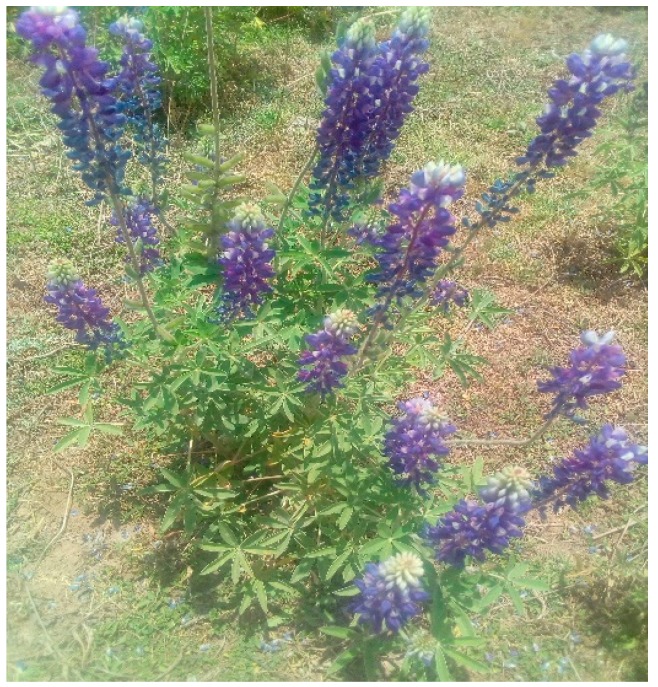
*Lupinus rotundiflorus* M.E. Jones. Photo: M. Ruiz.

**Figure 9 nutrients-11-01785-f009:**
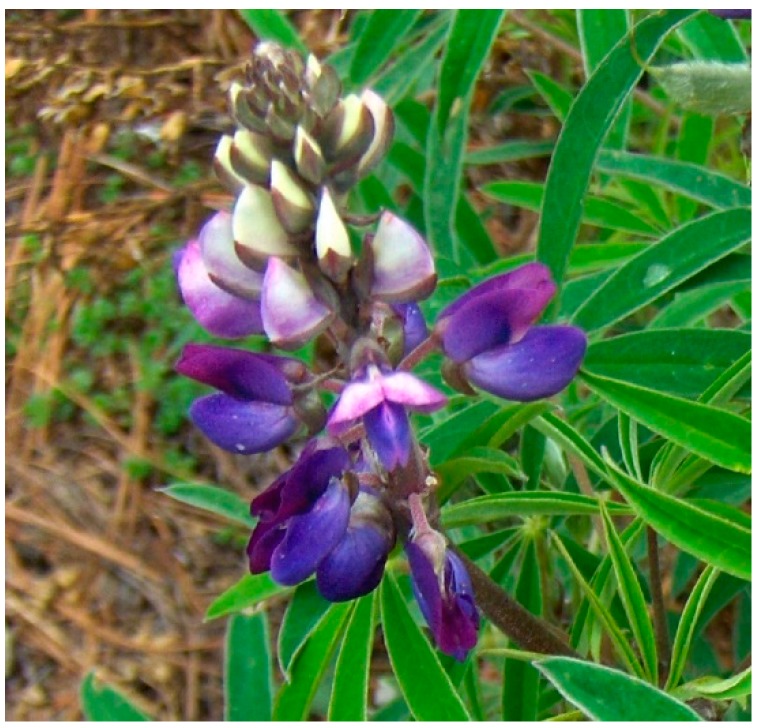
*Lupinus aschenbornii* S. Schauer. Photo: J. Ruiz.

**Figure 10 nutrients-11-01785-f010:**
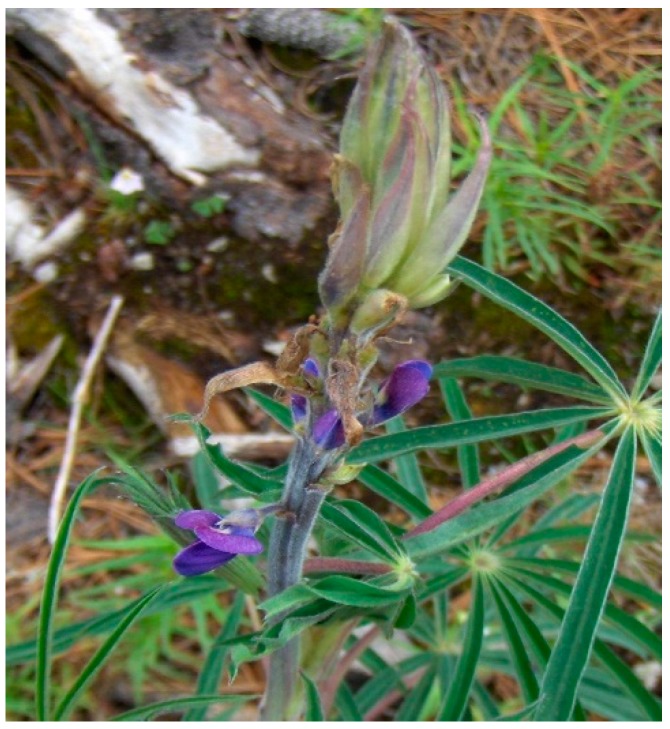
*Lupinus montanus* Kunth ssp. montanus. Photo: J. Ruiz.

**Figure 11 nutrients-11-01785-f011:**
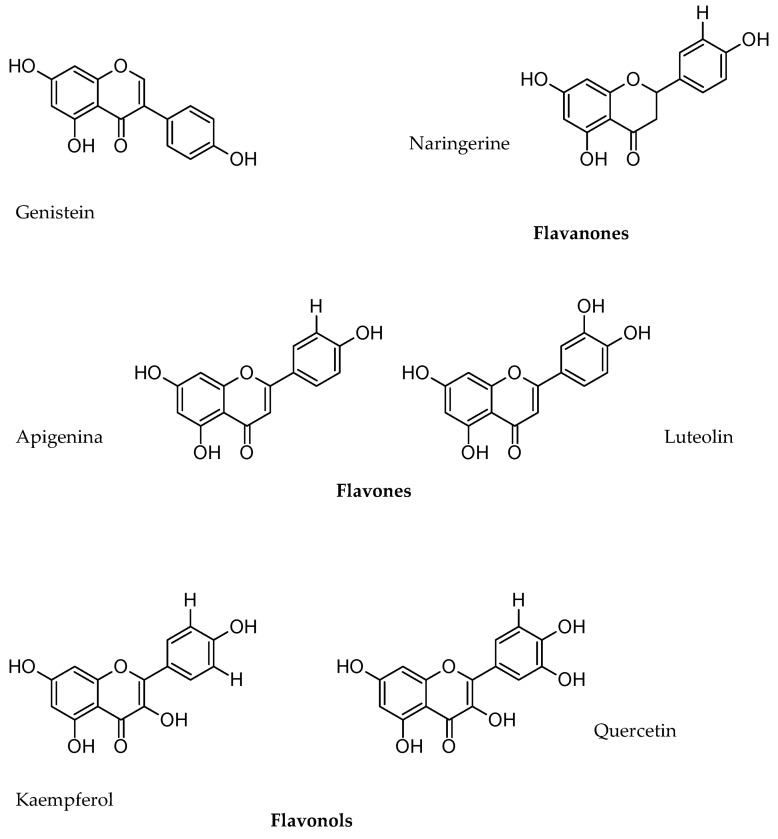
Genistein, flavones, flavonols, flavanones.

**Figure 12 nutrients-11-01785-f012:**
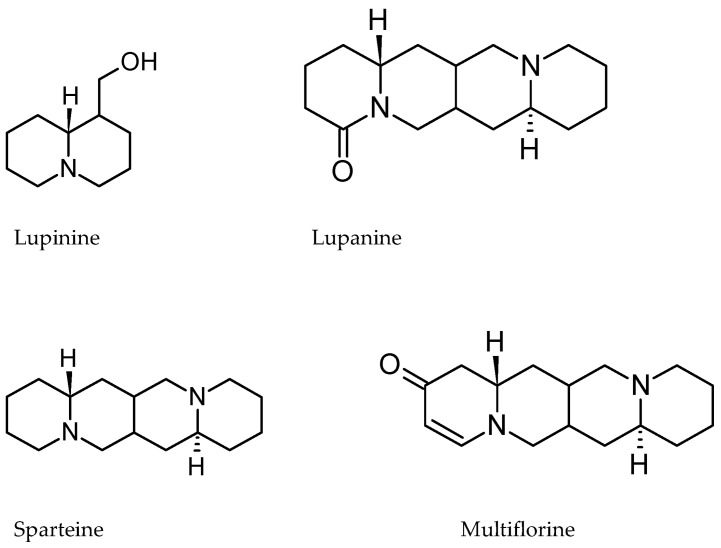
Main alkaloids of lupin species.

**Table 1 nutrients-11-01785-t001:** Chemical composition and alkaloids in the seeds of four lupine species (g/100 g DM).

Component	*L. albus*	*L. angustifolius*	*L. luteus*	*L. mutabilis*
Dry matter	90.4	90.6	91.7	62.0
Proteins	36.3	33.0	46.5	44.7
Ashes	3.9	3.7	3.7	3.0
Crude fat	11.5	6.8	4.6	14.07
Crude fiber	14.4	14.0	13.9	7.04
Alkaloids	0.04	0.06	0.1	1.27

Source: [6,7].

**Table 2 nutrients-11-01785-t002:** Fatty acid composition (g/kg DM) in different lupine species.

Fatty Acid	*L. albus*	*L. luteus*	*L. angustifolius*
16:0	78	48	110
18:0	16	25	382
18:1	530	210	382
18:2	172	475	371
18:3	95	75	53
20:0	55	45	12
22:0	58	79	19
22:1	58	79	19
n3/n6	0.55	0.16	0.14

Source: Reference [8].

**Table 3 nutrients-11-01785-t003:** Mineral Composition of Lupin Seed Flours (mg/100 g of Dry Matter).

Lupines	Ca	Mg	P	Cu	Fe	Mn	Zn
*L. albus* var. multolupa	139.0 ± 2.72	145.0 ± 1.65	332.1 ± 5.12	0.72 ± 0.06	3.80 ± 0.05	90.1 ± 1.34	4.30 ± 0.09
*L. albus* var. marta	133.6 ± 0.53	193.2 ± 0.86	468.1 ± 1.76	0.75 ± 0.03	6.20 ± 0.06	35.0 ± 0.32	5.24 ± 0.02
*L. angustifolius* var. troll	161.8 ± 1.49	191.9 ± 1.00	543.3 ± 2.25	1.02 ± 0.11	4.15 ± 0.03	7.6 ± 0.06	3.65 ± 0.03
*L. angustifolius* var. emir	143.0 ± 1.05	219.1 ± 1.25	613.4 ± 2.05	0.95 ± 0.05	4.26 ± 0.03	8.4 ± 0.04	3.79 ± 0.02
*L. luteus* var. 4486	134.8 ± 0.61	294.0 ± 2.39	715.5 ± 1.39	1.10 ± 0.03	5.84 ± 0.03	5.6 ± 0.03	5.90 ± 0.02
*L. luteus* var. 4492	110.4 ± 0.66	308.8 ± 1.84	845.7 ± 2.72	1.25 ± 0.04	7.05 ± 0.05	6.8 ± 0.02	6.42 ± 0.02

Source: [19].

**Table 4 nutrients-11-01785-t004:** Oligosaccharide content of lupine cultivars (g/100 g DM).

Lupine cultivar	Sucrose	Raffinose	Stachyose	Verbascose	Total
*Lupinus albus* var. marta	0.73 ± 0.0	5.5 ± 0.01	1.1 ± 0.0		
*Lupinus albus* var. multolupa	0.63 ± 0.0	5.9 ± 0.01	1.2 ± 0.03		
*L. albus* cv. multolupa	2.58 ± 0.06	0.62 ± 0.03	5.7 ± 0.06	0.19 ± 0.10	7.56 ± 0.10
*L. albus* cv. marta	3.09 ± 0.08	0.33 ± 0.02	7.2 ± 0.11	0.94 ± 0.01	8.51 ± 0.13
*L. albus* LO-3844	2.16 ± 0.20	0.44 ± 0.03	7.2 ± 0.31	ND	7.71 ± 0.33
*L. albus* LO-3846	3.13 ± 0.06	0.54 ± 0.02	6.85 ± 0.03	ND	7.39 ± 0.03
*L. albus* LO-3848	2.41 ± 0.19	0.36 ± 0.02	5.71 ± 0.57	ND	6.07 ± 0.55
*L. albus* LO-3855	2.84 ± 0.09	0.48 ± 0.02	4.98 ± 0.22	ND	5.46 ± 0.24
*L. luteus* LO-4486	1.38 ± 0.13	0.56 ± 0.11	7.01 ± 1.22	3.54 ± 0.37	11.1 ± 1.69
*L. luteus* LO-4492	1.21 ± 0.04	0.64 ± 0.04	8.61 ± 0.20	3.04 ± 0.02	12.3 ± 0.20
*L. luteus* LO-4500	1.01 ± 0.09	0.54 ± 0.03	6.13 ± 0.31	2.79 ± 0.14	9.46 ± 0.41
*L. angustifolius* LO-4817	5.05 ± 0.22	1.24 ± 0.09	5.11 ± 0.36	2.48 ± 0.07	8.82 ± 0.43
*L. angustifolius* LO-4820	2.55 ± 0.24	0.89 ± 0.07	3.62 ± 0.06	0.79 ± 0.03	5.30 ± 0.12
*L. angustifolius* LO-4822	2.91 ± 0.19	1.15 ± 0.07	5.19 ± 0.14	1.36 ± 0.01	7.70 ± 0.11
*L. angustifolius* cv. zapaton	3.16 ± 0.17	0.63 ± 0.04	4.52 ± 0.18	1.39 ± 0.13	6.54 ± 0.34

Source: [4,28].

**Table 5 nutrients-11-01785-t005:** The content of total phenolic compounds (gallic acid equivalent) and flavonoids (catechin equivalent) from seeds of lupine varieties.

Sample	Cultivar	Total Phenolic Compounds (mg GAE/100 g DM)	Total Flavonoids (µg Catechin/g DM)
*Lupinus albus*	Butan	212.12 ± 2.24	
	Boros	271.25 ± 3.75	
	Multolupa		1100 ± 17.6
*Lupinus luteus*	Lord	249.32 ± 4.72	
	Parys	317.88 ± 2.69	
*Lupinus angustifolius*	Bojar	269.72 ± 9.97	
	Zeus	258.42 ± 7.21	
	Troll		133 ± 12.6
	Emir		362 ± 9.00

Source: [30,34].

**Table 6 nutrients-11-01785-t006:** Content of phenols and phenolic acids in *Lupinus* species (mg/kg DW).

Sample	*Lupinus albus*		*Lupinus luteus*		*Lupinus angustifolius*	
	Butan	Boros	Lord	Parys	Bojar	Zeus
Apigenin-6,8-di-C-b-glucopyranoside (mg/100 g DM)	11.90 ± 0.33	14.3 ± 0.33	53.63 ± 0.44	63.14 ± 0.14	30.25 ± 0.22	27.78 ± 0.07
Gallic acid (mg/kg DM)	3.53 ± 0.24	3.43 ± 0.1	3.50 ± 0.33	4.21 ± 0.27	0.63 ± 0.01	0.62 ± 0.04
Protocatechuic (mg/kg DM)	12.96 ± 0.14	14.69 ± 0.36	35.90 ± 0.54	73.60 ± 1.71	12.50 ± 0.39	13.77 ± 0.39
p-Hydroxybenzoic (mg/kg DM)	22.77 ± 0.30	27.82 ± 0.68	1.06 ± 0.11	2.24 ± 0.21	43.73 ± 0.48	42.73 ± 0.31
Caffeic acid (mg/kg DM)	0.58 ± 0.05	0.09 ± 0.01	1.02 ± 0.06	1.22 ± 0.11	0.84 ± 0.09	0.56 ± 0.07
p-Coumaric acid (mg/kg DM)	0.11 ± 0.01	0.18 ± 0.01	0.03 ± 0.01	0.68 ± 0.14	0.42 ± 0.08	0.34 ± 0.06
Total (mg/kg DM)	39.96 ± 0.15	46.23 ± 1.22	41.52 ± 0.68	82.06 ± 1.52	58.14 ± 1.02	58.03 ± 0.87

Source: [30].

**Table 7 nutrients-11-01785-t007:** Antioxidant capacity of lupine seed extracts.

Sample	Cultivar	DPPH (mM Trolox) Eq/g	TRAP (mMTrolox) Eq/g
*Lupinus albus*	Butan	3.51 ± 0.2	0.33 ± 0.01
	Boros	6.78 ± 0.28	0.71 ± 0.01
*Lupinus luteus*	Lord	9.03 ± 0.33	1.44 ± 0.02
	Parys	8.12 ± 0.10	0.96 ± 0.07
*Lupinus angustifolius*	Bojar	6.89 ± 0.35	1.60 ± 0.05
	Zeus	7.47 ± 0.29	1.78 ± 0.04

Source: Reference [30]. DPPH, 2,2-diphenyl-1-picrilhydrazil; TRAP, total peroxil radical-trapping potential.

**Table 8 nutrients-11-01785-t008:** Proximate composition of seeds from species of Mexican wild lupins (g /100 g DM).

Component	Ash	Lipids	Crude Fiber	Crude Protein (*n* = 6.25)	Carbohydrates
*L. elegans*	4.2	5.8–7.3	12.9	43.6–45.4	31.7
*L. exaltatus*	3.4–5.2	5.8–8.7	14.6–27.0	32.1–43.9	22.9–32.8
*L. reflexus*	3.6–7.2	6.6–7.9	15.2–16.6	37.3–38.8	32.1–34.6
*L. rotundiflorus*	3.1–4.1	5.5–6.4	151	41.9–42.8	32.5
*L. simulans*	3.6	6.3	14.4	40.7	35.0
*L. splendens*	3.3–4.3	8.9–13.0	12.7–16.4	34.1–37.2	32.1–38.1
*L. mexicanus*	3.8–4.1	6.1–8.0	16.8	34.7–36.7	34.3
*L. madrensis*	3.5	6.8	15.4	41.4	32.8
*L. campestris*	4.4	7.5		40.5	39.3
*L. montanus*	3.6–4.3	7.1–10.0	26.5	42.4–45.9	28.3
*L. hintonni*	6.3	7.0		32.5	24.4

Source: [37,40,41,42,43,44].

**Table 9 nutrients-11-01785-t009:** Approximate chemical composition of protein isolates of *Lupinus* species from Mexico (g/100 g on dry base sample).

	Protein	Lipids	Ash	Fiber	Carbohydrates	Alkaloids
*L. campestris*	93.2	--	2.4	0.5	3.9	0.005
*L. exaltatus*	95.0	0.57	2.5	0.0	1.97	0.09
*L. elegans*	95	0.6	1.2	0.0	1.2	0.003

Sources: [38,45,46].

**Table 10 nutrients-11-01785-t010:** Saturated and unsaturated fatty acid content (g/kg) of wild lupin, collected at several locations of Jalisco and Zacatecas states in 1996–1997.

State Location	*L. exaltatus*, Jalisco			*L. montanus*, Jalisco			*L. stipulatus*, Zacatecas
	CG ^1^	CG ^2^	Z ^3^	T ^4^	T ^5^	B ^6^	ME ^7^
C14:0	2	2	2	8	9	7	8
C16:0	200	202	206	259	257	273	264
C18:0	48	56	66	36	37	69	76
C20:0	-	-	-	-	-	-	-
C22:0	-	-	-	23	22	23	25
C16:1	2	2	1	7	6	7	7
C18:1	138	124	138	96	99	171	137
C18:2	534	544	520	430	423	324	370
C18:3	76	71	67	142	146	126	113
C22:1	-	-	-	-	-	-	-
n6/n3	7/1	7.5/1	7.7/1	3/1	2.9/1	2.6/1	3.2/1
Ratio I/S	3.0	2.8	2.6	2.1	2.1	1.7	1.7

^1^ Ciudad Guzmán (1996), ^2^ Ciudad Guzmán (1997), ^3^ Zapopan (1996), ^4^ El Refugio, Tonila (1996), ^5^ El Refugio, Tonila (1997), ^6^ Bolaños (1997), ^7^ Monte Escobedo (1997). Source: Reference [47].

**Table 11 nutrients-11-01785-t011:** Dietary fiber (DF) content in five Mexican species of lupines.

Species	Dietary Fiber
*L. exaltatus*	17.72 ± 0.1
*L. elegans*	21.07 ± 0.0
*L. mexicanus*	20.90 ± 0.5
*L. montanus*	24.63 ± 0.1
*L. rotundiflorus*	27.93 ± 0.1

Source: [49].

**Table 12 nutrients-11-01785-t012:** Mineral content in wild lupine seeds (mg/kg in dry basis).

Species	Ca	P	Mg	Fe	Zn	Cu
*L. exaltatus*	1600–2052	584–5600	2300–2330	61.8–81.3	46.97–89.6	74.4–184
*L. elegans*	1777	6441	2656	70.9	73.6	64.8
*L. mexicanus*	3252	5865	2651	63.1	73.7	70.8
*L. montanus*	800–2074	6500–7690	2443–3000	77.7–73.7	46.87–73.3	56.2–89.7
*L. rotundiflorus*	1887	6166	2213	82.8	79.0	64.9
*L. campestris*	1200	6300	2300	70.74	46.97	93.3
*L. hintonii*	1000	5400	2600	49.39	38.18	67.7

Source: [49,50].

**Table 13 nutrients-11-01785-t013:** Oligosaccharide content in Mexican lupine seeds (mg/g of a dry sample).

Species	Sucrose	Raffinose	Stachyose	Verbascose	Totals
*L. campestris*	21.5 ± 0.76	11.7–13.65	57.16 ± 0.95	19.45 ± 0.59	90.26 ± 0.73
*Lupinus elegans*		7.82 ± 0.42	25.50 ± 1.21	37.20 ± 2.08	70.50 ± 3.71
*Lupinus montanus*		14.55 ± 0.92	37.32 ± 2.52	12.44 ± 0.94	64.31 ± 4.38
*Lupinus rotundiflorus*		5.60 ± 0.34	19.34 ± 0.82	29.57 ± 1.87	54.50 ± 3.03

Source: [53,54,55].

**Table 14 nutrients-11-01785-t014:** Quinolizidine alkaloid content in seeds from wild Mexican lupines (mg of alkaloids/g DM).

*Lupinus species*	Sparteine	Lupanine	3-hydroxy lupanine	13-hydroxy lupanine	Multiflorine	Hydroxyaphylline	Hydroxyaphyllidine
*L. exaltatus*	0.03	1.47–5.83	1.53	0.015	0.004		
*L. elegans*	nd	0.03	3.73	nd			
*L. splendens*	0.29	0.89	1.05	1.00			
*L. reflexus*	26.63	2.91	0.16	0.08			
*L. rotundiflorus*	0.11	11.50	4.19	nd			
*L. simulans*	0.40	8.87	2.76	0.09			
*L. madrensis*	0.02	10.63	2.08	0.03			
*L. montanus*	3.97	1.65		0.10	0.07		
*L. stipulatus*	0.04	0.1		0.12	0.2		
*L. mexicanus*		5.05–21.2		0.015	0.096–1.09		
*L. campestris*		1.2				498.0	1475.0

Source: [40,54,61,64].

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
