# Peer review of "Nutritional and Bioactive Compounds in Mexican Lupin Beans Species: A Mini-Review"

_nutrients, 2019, doi:10.3390/nu11081785_

Round 1
Reviewer 1 Report
In my opionion this paper by Ruiz Lopez et al. Is interesting, nice to read and may be suitable for publication. I suggest considering some minor comments that I attach below.
Introduction: some quite generic sentences about plants are present: I would suggest to focus more tightly on the topic.
Figure 1 may be too big, especially concerning font size.
I would suggest consider grouping Figures 2-5 to give the paragraph a tidier look.
Figure 6: please resize the molecule. Moreover, it is not the same style of Fig.11.
Please modify Fig.11: it has a bad resolution and a non-consistent style.
I would suggest removing Fig. 12, as it does not present original data.
Reviewer 2 Report
The language of the manuscript must be formal.
The grammatical issues must be checked and corrected.
The whole manuscript should be double checked for grammatical issues and fluency.
The quality of the 3D structures must be improved. The authors can not copy the structures from other sources.
It is recommended the 3D structures be drawn by Chemo-sketch or further similar softwares.
Very recent and high quality references are recommended to be cited. The following references are advised strongly to be cited in relevant areas.
1. Vahid Farzaneh, Jorge Gominho, Helena Pereira, Isabel S. Carvalho. Screening of the Antioxidant and Enzyme Inhibition Potentials of Portuguese Pimpinella anisum L. Seeds by GC-MS. Food Analytical Methods. https://doi.org/10.1007/s12161-018-1250-x. 2018.
2. Vahid Farzaneh, Isabel S. Carvalho. Modelling of Microwave Assisted Extraction (MAE) of Anthocyanins (TMA). Journal of Applied Research on Medicinal and Aromatic Plants, https://doi.org/10.1016/j.jarmap.2017.02.005. 2017.
3. Vahid Farzaneh, Isabel S. Carvalho. A review of the health benefit potentials of herbal plant infusions and their mechanism of actions. Published in Industrial crops and products. https://doi.org/10.1016/j.indcrop.2014.10.057. 2015.
Look forard to receiving the revised version of the manuscript.
Reviewer 3 Report
7-July--2019
Journal: Nutrients
Title: Nutritional and bioactive compounds in Mexican lupin beans species. A review
Authors: Mario Alberto Ruiz-López, Lucia Barrientos-Ramirez, Pedro Macedonio García-López, Elia Herminia Valdez-Miramontes, Francisco Zamora-Natera, Ramón Rodriguez-Macias, Eduardo Salcedo-Pérez, Jacinto Bañuelos-Pineda and Jesús Vargas-Radillo
Dear Editor,
The authors have investigated the Mexican lupin beans species. The authors compared all species in relation to each other, using data and structures from previous scientific papers and tables. Unfortunately, the originality is missing as there aren`t biological activity experiments and clinical trials evidences included. The manuscript is in need of thorough revision and reorganization, and in addition it shows high plagiarism percent.
Taken together, the review carries the scientific merit but need a lot of work to be accepted for publication (though after major revision). Hereby, we are enlisting some few points for consideration when amending the manuscript.
Comments to Authors:
Abstract:
1. Authors should provide high quality graphical abstract in accordance to journal instruction.
2. Authors would add properties of domesticated and semi-domesticated species of the genus Lupinus.
3. Please rewrite the abstract with more focus on the common bioactive compounds and biological activities of these species.
Key words:
1. Authors would double check on (bioactives) as key words.
2. Authors would select more specific key words.
General comments:
1. Please put genus name in italic form as Lupins.
2. Author could include some additional references related to this review script.
3. Author may include the future prospective of nutritional, nutraceutical and pharmacological sectors.
4. Please add the nutritional applications of Mexican lupin beans species.
5. Fig.1, 6 and 11. Please make them with high resolution and clear structure (avoid copy and paste).
6. Authors would add more information to enrich the introduction.
7. Authors would mention the technical analysis used for preparing these molecules.
8. Authors would add medicinal importance of low gluten foods from these species.
9. Please add more knowledge about this technique (TRAP technique).
10. Could you determine n3/n6 ratio in page 12.
11. Authors could check the writing style through out of the manuscript.
12. Could you add the clinical trials for these species?
13. The critical thinking and analysis of the collected data is missing (what is new? And what could be added to the reader).
14. In figure 12. Results of alkaloids MS spectra of these species may be different in accordance to the experiment conditions.
15. The authors could benefit from the following reference in the introduction:
Boldootar, D., Hellman, B., Goransson, U. and El-Seedi, H.R. (2018): Novel alkaloid from the Mongolian medicinal plant: Leptopyrum fumarioides (L.) Rchb. In preparation.
References:
Authors would check the style of writing references in accordance to journal instruction.
Please consider rewriting this manuscript as it seems to contain noticeable high plagiarism.
Round 2
Reviewer 3 Report
20-July--2019
Journal: Nutrients
Title: Nutritional and bioactive compounds in Mexican lupin beans species. A review
Authors: Mario Alberto Ruiz-López, Lucia Barrientos-Ramirez, Pedro Macedonio García-López, Elia Herminia Valdez-Miramontes, Francisco Zamora-Natera, Ramón Rodriguez-Macias, Eduardo Salcedo-Pérez, Jacinto Bañuelos-Pineda and Jesús Vargas-Radillo
Dear Editor,
Taken together, the review carries the scientific merit but need also some of work to be accepted for publication (though after minor revision). Hereby, we are enlisting some few points for consideration when amending the manuscript. The manuscript is in need of thorough revision and reorganization.
Comments to Authors:
1. Authors would mention the technical analyses used for isolation these molecules from the Mexican lupin beans species.
2. Please add the nutritional applications of Mexican lupin beans species in a separated section.
3. Author may include the future prospective of nutritional, nutraceutical and pharmacological sectors in more details.
4. The authors could benefit from the following reference in the introduction:
El-Seedi, Hesham R., Zahra, M.H., Göransson, U. and Verpoorte, R. (2007): Cyclopeptide alkaloids. Phytochemistry Reviews 6, 143-165.
Author Response
Comments to reviewer:
1. Authors would mention the technical analyses used for isolation these molecules from the Mexican lupin beans species.
2. Please add the nutritional applications of Mexican lupin beans species in a separated section.
3. Author may include the future prospective of nutritional, nutraceutical and pharmacological sectors in more details.
4. The authors could benefit from the following reference in the introduction:
El-Seedi, Hesham R., Zahra, M.H., Göransson, U. and Verpoorte, R. (2007): Cyclopeptide alkaloids. Phytochemistry Reviews 6, 143-165.
RESPONSE 1: The techniques used for the isolation and analysis of the molecules are mentioned in each section
RESPONSE 2: In a separate section we adding nutritional application of Mexican lupin species, before of conclusions
RESPONSE 3: We included future perspective of nutritional, nutraceutical and pharmacological sector of Mexican lupin bean species.
RESPONSE 4:The follow reference was included:
El-Seedi, Hesham R., Zahra, M.H., Göransson, U. and Verpoorte, R. (2007): Cyclopeptide alkaloids. Phytochemistry Reviews 6, 143-165.
